# Increased Susceptibility of *Mycobacterium tuberculosis* to Ethionamide by Expressing PPs-Induced Rv0560c

**DOI:** 10.3390/antibiotics11101349

**Published:** 2022-10-04

**Authors:** Hoonhee Seo, Sukyung Kim, Hafij Al Mahmud, Omme Fatema Sultana, Youngkyoung Lee, Youjin Yoon, Md Abdur Rahim, Sujin Jo, Jiwon Choi, Saebim Lee, Ho-Yeon Song

**Affiliations:** 1Department of Microbiology and Immunology, School of Medicine, Soonchunhyang University, Cheonan-si 31151, Korea; 2Probiotics Microbiome Convergence Center, Soonchunhyang University, Asan-si 31538, Korea; 3College of Pharmacy, Dongduk Women’s University, Seoul 02748, Korea

**Keywords:** *Mycobacterium tuberculosis*, drug-resistant, PPs, Rv0560c, ethionamide (ETH) booster

## Abstract

Tuberculosis, an infectious disease, is one of the leading causes of death worldwide. Drug-resistant tuberculosis exacerbates its threat. Despite long-term and costly treatment with second-line drugs, treatment failure rates and mortality remain high. Therefore, new strategies for developing new drugs and improving the efficiency of existing drug treatments are urgently needed. Our research team reported that PPs, a new class of potential anti-tuberculosis drug candidates, can inhibit the growth of drug-resistant *Mycobacterium tuberculosis*. Here, we report a synergistic effect of PPs with ethionamide (ETH), one of the second-line drugs, as a result of further research on PPs. While investigating gene expression changes based on microarray and 2DE (two-dimensional gel electrophoresis), it was found that PPs induced the greatest overexpression of Rv0560c in *M. tuberculosis*. Based on this result, a protein microarray using Rv0560c protein was performed, and it was confirmed that Rv0560c had the highest interaction with EthR, a repressor for EthA involved in activating ETH. Accordingly, a synergistic experiment was conducted under the hypothesis of increased susceptibility of ETH to *M. tuberculosis* by PPs. As a result, in the presence of 0.5× MIC PPs, ETH showed a growth inhibitory effect on drug-sensitive and -resistant *M. tuberculosis* even at a much lower concentration of about 10-fold than the original MIC of ETH. It is also suggested that the effect was due to the interaction between PPs and Rv2887, the repressor of Rv0560c. This effect was also confirmed in a mouse model of pulmonary tuberculosis, confirming the potential of PPs as a booster to enhance the susceptibility of *M. tuberculosis* to ETH in treating drug-resistant tuberculosis. However, more in-depth mechanistic studies and extensive animal and clinical trials are needed in the future.

## 1. Introduction

Tuberculosis, a communicable disease caused by *Mycobacterium tuberculosis*, is one of the leading causes of death worldwide, killing 1.3 million HIV-negative (up from 1.2 million in 2019) and 214,000 HIV-positive people in 2020 (up from 209,000 in 2019) [1]. Moreover, the development of drug resistance, including multidrug resistance (MDR), extensive drug resistance (XDR), and total drug resistance (TDR), in *M. tuberculosis* decisively impedes the efficacy of currently available drug therapies [2]. To respond to the threat of tuberculosis to humankind, it is necessary to develop new drugs and treatment strategies based on the basic molecular biology of tuberculosis.

The development of anti-tuberculosis drugs started with the development of streptomycin in 1944 by Nobel Prize laureate Selman Waksman, followed by the development of *p*-aminosalicylic acid in 1949, isoniazid in 1952, pyrazinamide in 1954, ethambutol in 1962, rifampin in 1963, and cycloserine in 1964 [3]. As mentioned above, the current anti-tuberculosis drugs discovered from the 1950s to the 1970s were developed through clinical trials until the 1980s. However, there was a discovery void of tuberculosis drug R&D for about 30 years until about 2000 [4]. The void in anti-tuberculosis drug discovery is underscored by the fact that it coincided with the emergence of drug-resistant *M. tuberculosis*. In contrast, the recent FDA approval of bedaquiline and delamanid for MDR/XDR tuberculosis spurs the development of tuberculosis drugs [5].

To develop antibiotic-resistant tuberculosis treatment, our research team discovered and reported anti-tuberculosis drug candidates called PPs (methyl (S)-1-((3-alkoxy-6,7-dimethoxyphenanthren-9-yl) methyl)-5-oxopyrrolidine-2-carboxylate derivatives) [6]. PPs are a new class of drugs with different basic structures from existing tuberculosis drugs. Both in vitro and in vivo experiments confirmed that PPs could effectively inhibit the growth of drug-resistant *M. tuberculosis*. PPs also showed no detectable toxicity in single/repeat oral toxicity and genotoxicity studies performed at the GLP (good laboratory practice) level. These results suggest that PPs have sufficient potential for developing new drugs for treating drug-resistant tuberculosis. Therefore, various experiments were additionally performed, and here, we report on gene expression by PPs and their excellent synergistic effect with ethionamide (ETH).

Briefly explaining the article preview, during the gene expression profiling experiment of *M. tuberculosis*, overexpression of Rv0560c by PPs was confirmed, and the function of this protein was explored. As a result of performing a protein microarray for *M. tuberculosis* using Rv0560c protein, it was confirmed that Rv0560c had the highest interaction with EthR, an inhibitor of EthA involved in ETH activation. Accordingly, in vitro and in vivo experiments were conducted with the hypothesis that the susceptibility of ETH to *M. tuberculosis* by PPs would increase. The results are described in this article.

## 2. Results

### 2.1. Exploring PPs-Induced Alterations of Gene Expression in M. tuberculosis

Gene expression changes in *M. tuberculosis* H37Rv after treatment with PP1S were investigated (Figure 1). A two-dimensional gel electrophoresis (2-DE) experiment was performed to examine changes in protein expression in *M. tuberculosis* treated with PP1S at 1× and 10× MICs (Figure 1A). As a result, seven differentially expressed proteins with fold changes of at least 1.5 between drug-treated cells and untreated samples were identified (Figure 1B). Of these proteins, the two most over-expressed proteins (spots 1 and 2) were identified as the same protein, Rv0560c, a putative benzoquinone methyltransferase. Two under-expressed proteins (spots 3 and 4) were identified as enoyl-CoA hydratase and translation elongation factor EF-Tu, respectively. Another three proteins (spots 5, 6, and 7) showed no consistent changes. In the results of comparative transcript analysis of *M. tuberculosis* after treatment with PP1S or PP2S through microarray experiments, it was also confirmed that the Rv0560c gene was the most highly upregulated (Figure 1C). Results of microarray experiments analyzed by genomic category are presented in Appendix A. Overexpression of Rv0560c was verified by real-time PCR (Figure 1D) and Western blotting experiments (Figure 1E). PPs upregulated Rv0560c in a concentration-dependent manner. Salicylates used as positive control drugs also upregulated Rv0560c.

### 2.2. ETH-Boosting Activity of PPs

Rv0560c, confirmed to be overexpressed by PPs, was synthesized. The interaction with all proteins of *M. tuberculosis* was investigated using this (Figure 2). Protein microarray experiments were performed on the interaction with all proteins of *M. tuberculosis*. The results show that EthR had the highest interaction with Rv0560c (Figure 2A). To verify this result, Rv0560c and EthR proteins were synthesized, and direct interaction between the two proteins was tested through surface plasmon resonance (SPR) experiments (Figure 2B). EthR induced a change in refractive index when it was added to Rv0560c immobilized on a sensor surface. The K_D_ of EthR was 31.9 μM, indicating that it could bind to Rv0560c at a low micromolar affinity.

Given the critical role of EthR in the repression of EthA, we tested whether PPs could boost the anti-tubercular activity of ETH (Figure 3). In the presence of 0.5× MIC of PP1S or PP2S, ETH inhibited the growth of *M**. tuberculosis* H37Rv even at concentrations more than 10-fold lower than the MIC of ETH alone (Figure 3A). However, no change in MIC was observed for PPs in the presence of ETH at 0.5× MIC, which is the opposite condition (Figure 3B). The effect of increasing the sensitivity of ETH in *M. tuberculosis* of PPs was similar to the results for the XDR strain (Appendix A). Treatment with salicylates used as positive controls also increased the sensitivity of ETH to *M. tuberculosis* H37Rv (Appendix A).

In the *M**. tuberculosis*-infected animal model, compared to the group administered with 10 mg/kg/day of ETH alone, mice treated with PP2S in combination with 10 mg/kg/day of ETH showed significantly lower numbers of *M**. tuberculosis* (Figure 4). In addition, PPs were tested for synergistic effects with first-line drugs, and there were no synergistic or antagonistic effects for all drugs tested (Table 1).

### 2.3. Interaction between Rv2887, a Repressor of Rv0560c, and PPs

It was hypothesized that the overexpression of Rv0560c might be possible through the interaction of Rv2887, a repressor of Rv0560c, with PPs. An experiment was conducted for this (Figure 5). The interaction between PPs and Rv2887 was analyzed using an in silico method (Figure 5A). PP1S and PP2S could dock into a hydrophobic cavity formed by residues Leu13, Leu20, and Leu35 of Rv2887, while phenanthrene rings of PPs could contact the Arg21 side chain through cation-π interactions. These results were also confirmed by SPR, demonstrating direct binding of PP1S and PP2S to Rv2887, with K_D_ values of 274 μM and 250 μM, respectively (Figure 5B).

## 3. Discussion

*M. tuberculosis* is a globally distributed lethal human pathogen. Its path adaptation is further enhanced by the modularity, flexibility, and interactivity of mycobacterial effectors and their regulators [7]. There is a need to develop new drugs and various therapies based on understanding the mechanisms for maintaining adaptability that characterize these mycobacteria.

Our research team reported the anti-tuberculosis effect of PPs, a new class of candidate drugs structurally different from existing tuberculosis drugs [6]. As reported in our previous study, results of anti-tuberculous activity and toxicity of PPs confirmed the feasibility of entering the next stage of development for PPs. Thus, we proceeded with various in-depth studies. While exploring the *M. tuberculosis* gene expression pattern, it was found that PPs significantly upregulated Rv0560c. Based on this finding, the synergistic effect of PPs with ETH was inferred and verified.

Two highly over-expressed proteins in response to PPs were identified as the same benzoquinone methyltransferase encoded by *Rv0560c*. Although the function of Rv0560c is currently unknown, it shares sequence identities with a benzoquinone methyltransferase involved in ubiquinone biosynthesis in *Escherichia* [8,9]. Rv0560c is also suggested to be involved in menaquinone biosynthesis [5,6,7], specifically in demethylmenaquinone-to-menaquinone conversion [8], the same as Rv0558 [8,9,10]. Our data confirmed that Rv0560c could bind strongly to EthR, a TetR/CamR family repressor that controls ETH bioactivation in *M. tuberculosis* [10]. Given the critical role of EthR in the repression of EthA (ETH bioactivator) [11], we tested whether PPs could boost the anti-tubercular activity of ETH. A synergistic effect of PPs at 0.5× MIC was demonstrated even at a concentration lower than 1× MIC of ETH. This synergistic effect was also seen in a tuberculosis mouse model. In an experiment using salicylate known to cause overexpression of Rv0560c [8], it was also confirmed that ETH was more sensitive to *M. tuberculosis*, validating the results of this study. Domain and homology analyses suggested that Rv2887 might be a transcriptional regulator [12]. Our data suggest that Rv0560c is controlled by a tight interaction between the repressor Rv2887 and its binding motif [9,13,14]. Overexpression of MarR (multiple antibiotic resistance repressor) family transcription factor Rv2887 could lead to repression of *Rv0560c* [13]. *Rv0560c* is upregulated following the deletion of *Rv2887* [14]. This study confirmed that PPs could bind to Rv2887, similar to previous reports showing that salicylate can increase the expression of Rv0560c by binding to Rv2887 [15]. Therefore, we extend the existing EthR-EthA pathway [16] and Rv2887-Rv0560c [13] and propose a new Rv2887-Rv0560c-EthR-EthA pathway in which the two existing pathways are merged.

Despite these results, this study still has limitations. As mentioned above, in ETH’s activation by EthA, how EthR acts as a repressor for EthA has already been well understood [11,16]. With this mechanism in mind, the observed interaction between Rv0560c overexpressed by PPs and EthR and the result of increased sensitivity of PPs to ETH was judged based on the EthR-EthA pathway mechanism. However, the lack of direct experimental results for this remains a limitation and needs to be investigated in more detail in future studies. In addition, the function of Rv2887 as a repressor for Rv0560c has been previously reported [9,13,14]. This study confirmed the interaction between PPs and Rv2887 and the result of overexpression of Rv0560c. However, more experiments are required to determine that this result is due to the Rv2887-Rv0560c pathway. Moreover, more extensive and in-depth animal validation is needed, along with various additional experiments on drug target validation.

ETH, one of the most effective second-line drugs, has several side effects. Thus, it is essential to reduce its dose while maintaining its anti-tuberculotic effect [17]. Accordingly, several research studies have been reported on developing new drugs that can increase the sensitivity of ETH to *M. tuberculosis* [17,18]. Therefore, the synergistic effect of PPs with ETH has significant implications for the treatment of tuberculosis. Further mechanistic studies, extensive animal efficacy, and clinical trials are needed in the future.

In conclusion, overexpression of Rv0560c by PPs increases the sensitivity of ETH to *M. tuberculosis* through the Rv2887-Rv0560c-EthR-EthA pathway newly proposed in this study (Appendix A). Therefore, using PPs could be a new treatment strategy for drug-resistant tuberculosis. Further research needs to be carried out extensively before PPs can be applied to clinical treatment.

## 4. Method

### 4.1. M. tuberculosis Strains

An H37Rv (ATCC 27294) strain of *M. tuberculosis* was purchased from the American Type Culture Collection (ATCC, Manassas, VA, USA). A clinically isolated XDR strain (KMRC 00203-00197) was purchased from the Korean Microorganism Resource Center (KMRC, Chungbuk, Korea). For broth culture of *M. tuberculosis*, 7H9 broth (BD Difco, Franklin Lakes, NJ, USA) supplemented with 10% OADC (BD Difco, Franklin Lakes, NJ, USA), 0.2% glycerol (Sigma, Saint Louis, MO, USA), and 0.25% tween80 (Sigma, Saint Louis, MO, USA) was used. *M. tuberculosis* was cultured in an incubator at 37 °C and 130 rpm. For agar plate culture, 7H10 agar (BD Difco, Franklin Lakes, NJ, USA) was supplemented with 10% OADC and 0.2% glycerol, and *M. tuberculosis* was cultured in an incubator at 37 °C.

### 4.2. Drugs

PPs were prepared as previously reported [6]. Among PPs, the structures of PP1S and PP2S tested in this study are presented in Appendix A. Isoniazid, rifampicin, streptomycin, ethambutol, pyrazinamide, ETH, and salicylate were purchased from Sigma-Aldrich (USA).

### 4.3. Exploration of Gene Expression of M. tuberculosis Changed by Drugs

*M. tuberculosis* culture medium in the intermediate exponential growth stage was used to search for gene expression in *M. tuberculosis* changed by drugs. A freshly prepared *M. tuberculosis* culture medium that had reached the intermediate exponential phase was adjusted to an OD600 nm 0.8 using a Spectrophotometer (DR1900, Hach, Loveland, CO, USA). Then, PPs were treated with 1× MIC and 10× MIC, respectively, and incubated for 6 h in an incubator at 37 °C and 180 rpm. The *M. tuberculosis* culture medium volume was 30 mL, and three samples were tested for each drug, including the control group not treated with PPs. The *M. tuberculosis* culture medium exposed to the drug was centrifuged at 3000 rpm for 10 min and washed to obtain a cell pellet. RNA was extracted from *M. tuberculosis* harvest and used for the microarray experiment. Extracted protein was used for the 2DE experiment. The microarray and 2DE experiment are described immediately below, but detailed methods are presented in Appendix A.

### 4.4. Microarray

Microarray experiments with drugs in *M. tuberculosis* were performed according to previously reported methods [19]. First, RNA was extracted from *M. tuberculosis* with an RNAprotect Bacteria Reagent kit (QIAGEN, Hilden, Germany). The extracted RNA was sent to ebiogen (Seoul, Korea) for microarray analysis. Briefly, RNA samples were reverse transcribed into cDNA, and target cRNA probes were synthesized and hybridized on a MYcroarray.com (accessed on 08 September 2022) (*M. tuberculosis*) 3 × 20 k Microarray. Images were then acquired with an Agilent DNA microarray scanner (Agilent, Technology, Santa Clara, CA, USA) and quantified using Feature Extraction software (Agilent Technology, Santa Clara, CA, USA). Differentially transcribed genes were sorted by functional category [20]. A hypergeometric distribution method was used to determine which specific functional category genes were affected by the drug [21,22]. Genes identified with a fold change cutoff of >2 and *p* < 0.01 were analyzed for an in-depth study.

### 4.5. 2DE

Control and drug-treated cells were harvested, lysed in lysis buffer (pH 7, 0.3% SDS, 28 mM Tris base), and transferred to tubes containing silica beads (0.1 mm, MP biomedicals, Irvine, CA, USA). These cells were disrupted with 10 pulses for 30 s using a BeadBug microtube homogenizer (Benchmark Scientific, Inc., Sayreville, NJ, USA) at 4000 Hz with medium ice-cooling for 20 s. The concentration of extracted protein was measured by BCA (bicinchoninic acid) protein assay (Pierce, Waltham, MA, USA). 2DE experiment was performed by ProteomeTech (Seoul, Korea) as previously described [6]. Protein spots were analyzed by ImageMaster 2D Platinum software (Amersham Biosciences, Amersham, United Kingdom). As reported, separation of proteins from the gel and protein identification were performed [23,24].

### 4.6. Confirmation of Overexpression of Rv0560c Gene Using Real-Time PCR and Western Blot

As previously reported, real-time PCR experiments for *M. tuberculosis* were performed [25]. The *Rv0560c* gene targeting primers used in this study were as follows: sense primer sequence (5′-3′), GTAGAACTGGCTCGGCATGA; antisense primer (5′-3′), CCGGTCGAATACCAACACGA. Western blotting experiments were also performed, as previously reported [26].

### 4.7. Protein Microarray

First, Rv0560c protein was biotinylated using a biotinylation kit (EZ-Link Micro NHS-PEG4, Thermo Fisher Scientific, Waltham, MA, USA). MTBprot™ *Mycobacterium tuberculosis* Proteome Microarray (BC Biotechnology, China) containing 4262 full-length recombinant *M. tuberculosis* proteins was treated with 3 µg of Rv0560c biotinylated sample protein for 8 h at 4 °C and sequentially incubated with Streptavidin-Alexa Fluor 546 conjugate. Rv0560c-specific binding proteins were detected by scanning with a GenePix 4100A microarray laser scanner (Molecular Devices, San Jose, CA, USA). This study was conducted by Geneonbiotech (Daejeon, Korea).

### 4.8. In Silico Protein Modeling and Docking

Molecular docking studies were performed with a Glide docking program of the Schrödinger suite to evaluate the binding mode of PPs within the Rv2887 protein (PDB code 5X7Z) using an OPLS2005 force field. The grid center was set as the center of the selected residue, and the length of one side of the cubic grid was 15 Å. After the grid was created, ligand molecules were docked to the generated acceptor grid using the Glide SP docking program.

### 4.9. SPR Analysis

The ProteOn XPR36 protein interaction array system (Bio-Rad, Hercules, CA, USA) interacted between Rv0560c protein-EthR protein and Rv2887 protein-PPs. After the purified Rv0560c protein or Rv2887 protein was immobilized on the ProteOn GLH sensor chip, dilutions of the test EthR protein and PPs were prepared using PBS at various concentrations and then passed over the chip at a flow rate of 100 μL/min. Data were analyzed with ProteOn Manager software. The rate of complex formation is denoted by the association constant (k_a_, M^−1^s^−1^), and the rate of complex decay is denoted by the given dissociation constant (k_d_, s^−1^), respectively.

### 4.10. In Vitro Evaluation of the Synergistic Anti-Tuberculosis Effect

The synergistic effect of drugs against *M. tuberculosis* was determined by previous reports [27,28]. It was evaluated against H37Rv and XDR *M. tuberculosis* based on the checkerboard titration method using 96-well plates. In each well, 200 μL of 7H9 broth was mixed with *M. tuberculosis* (1.6 × 10^5^ CFU/mL), and various concentrations of each drug were determined by the 2-fold serial dilution method. After 7 days of incubation in aerobic conditions at 37 °C, 20 μL of 0.02% resazurin solution (Sigma, Saint Louis, MO, USA) was added to each well and incubated at 37 °C for an additional 2 days. Wells with the growth of *M. tuberculosis* turned pink, while wells with *M. tuberculosis* growth suppressed remained blue, the original color of resazurin. The color was measured and quantitatively analyzed with a Victor multimode microplate reader (PerkinElmer, Waltham, MA, USA). The MIC was the lowest drug concentration inhibiting *M. tuberculosis* growth. The FICI (fractional inhibitory concentration index) was calculated using these determined MIC values.

### 4.11. Evaluation of ETH-Boosting Effect in a Pulmonary Tuberculosis Mouse Model

In this experiment, BALB/c female mice (Dooyeol biotech, Seoul, Korea) were used (6 mice per group). A freshly cultured *M. tuberculosis* H37Rv culture solution with an OD_600 nm_ of about 0.8 was sprayed for 30 min using a nebulizer (Omron, Kyoto, Japan). A low dose of about 2 log CFU per mouse was used to infect the mouse. The drug was dissolved in corn oil (Sigma, Saint Louis, MO, USA). Mice were treated with the drug once a day, 5 days a week, for 4 weeks from the first day of infection. These mice were then sacrificed, and their lung bacterial load was enumerated as CFU. This animal study was conducted under the Soonchunhyang University Institutional Animal Care and Use Committee (IACUC, SCH22-0098).

### 4.12. Synthesis of Proteins and Preparation of Polyclonal Antisera

Rv3855 (Appendix A), Rv2887 (Appendix A), and Rv0560c (Appendix A) proteins were synthesized, and rabbit polyclonal antisera against Rv0560c protein was prepared (Appendix A). Experiments on these were performed by Young In Frontier (Seoul, Korea), and detailed methods are presented in Appendix A.

## Figures and Tables

**Figure 1 antibiotics-11-01349-f001:**
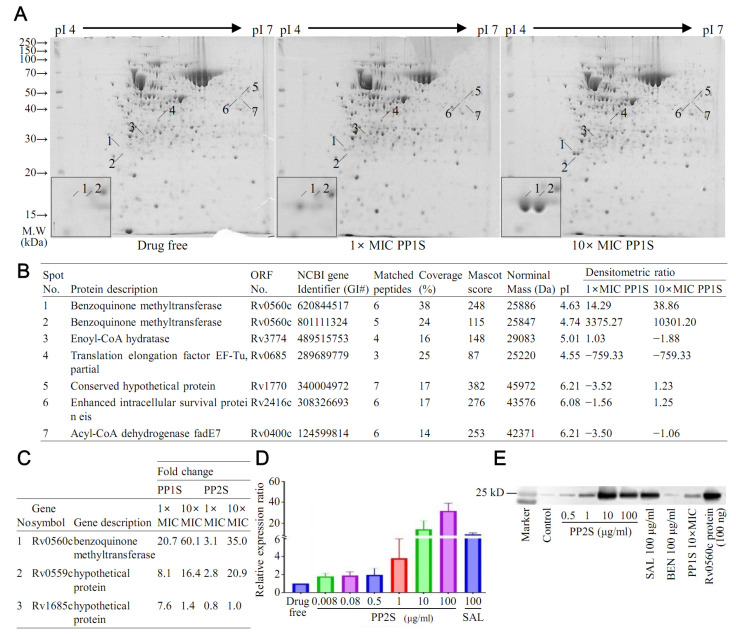
Gene expression changes in PPs-treated *M. tuberculosis* H37Rv. (**A**) Two-dimensional gel electrophoresis (2DE) analysis for protein expression in *M. tuberculosis* after treatment with 1× MIC and 10× MIC PP1S at 37 °C for 6 h. (**B**) Protein spots showing significantly different expressions between PP1S treated and untreated groups were identified. Two spots whose expressions were significantly increased by PP1S were found to be Rv0560c. (**C**) As a result of confirming the gene transcription pattern through microarray under the same conditions as 2DE, the most upregulated gene by PPs was Rv0560c. Overexpression of Rv0560c by PPs was confirmed by (**D**) RT-PCR and (**E**) Western blotting using an Rv0560c-specific antibody. (**D**,**E**) Rv0560c gene was also upregulated by salicylate (SAL). RT-PCR results are expressed as mean ± standard deviation.

**Figure 2 antibiotics-11-01349-f002:**
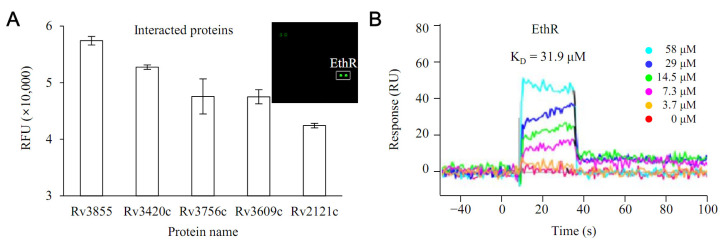
Interaction between Rv0560c protein and EthR protein. (**A**) A protein microarray experiment using biotinylated Rv0560c protein confirmed that Rv3855 (EthR) had the most potent interaction with Rv0560c protein among all proteins of *M. tuberculosis*. (**B**) The interaction of EthR with Rv0560c was verified through SPR (surface plasmon resonance).

**Figure 3 antibiotics-11-01349-f003:**
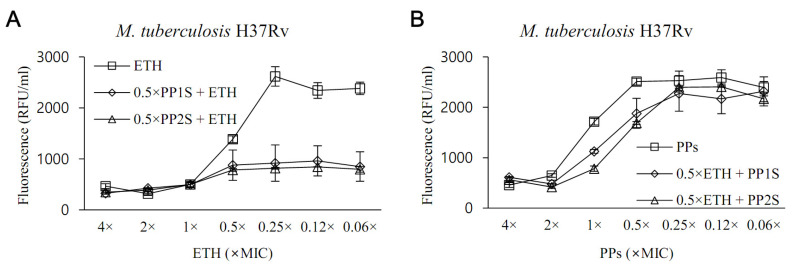
In vitro evaluation of the effect of PPs on the susceptibility of *M. tuberculosis* to ethionamide (ETH). Whether PPs could increase the anti-tuberculous activity of ETH was tested against the H37Rv strain. (**A**) In the presence of PP at 0.5× MIC, ETH significantly inhibited the growth of *M. tuberculosis* over concentrations ranging from 0.5× to 0.06× lower than 1× MIC (*p* < 0.05). (**B**) Conversely, no significant susceptibility-increasing effect of *M. tuberculosis* to ETH was observed at the various concentrations of PPs tested in the presence of 0.5× MIC of ETH (*p* > 0.05). Values were expressed as mean ± standard deviation, and statistical significance was determined using an unpaired Student’s *t*-test by comparing fluorescence values of samples treated with either ETH or PPs alone.

**Figure 4 antibiotics-11-01349-f004:**
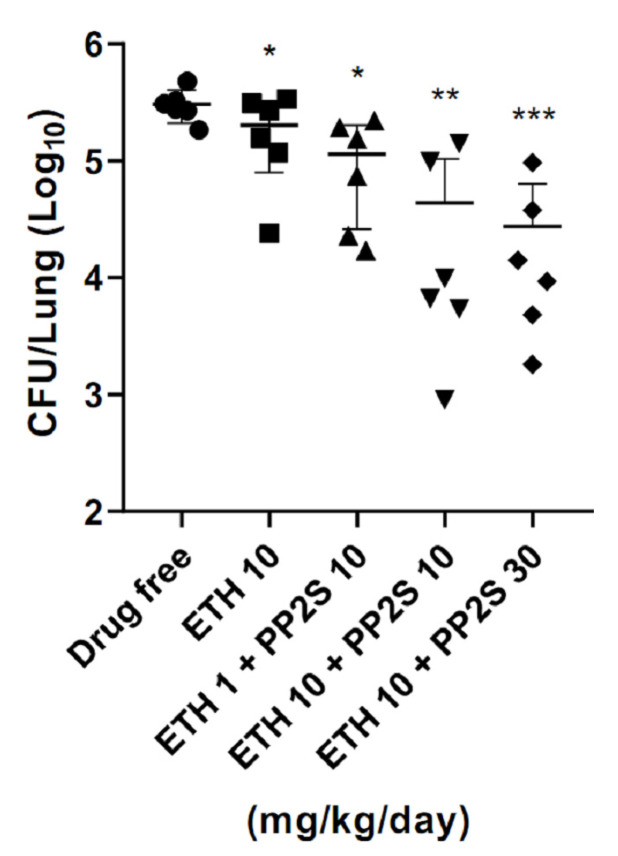
Synergistic effect of PPs and ethionamide (ETH) in the pulmonary tuberculosis mouse model. In the mouse model infected with *M. tuberculosis*, the load of *M. tuberculosis* in the lungs was lower in the group administered with PP2S and ETH in combination than in the group administered with ETH alone. Data are expressed as mean ± standard deviation. Student’s *t*-test was used to compare groups (* *p* < 0.05, ** *p* < 0.01, *** *p* < 0.001).

**Figure 5 antibiotics-11-01349-f005:**
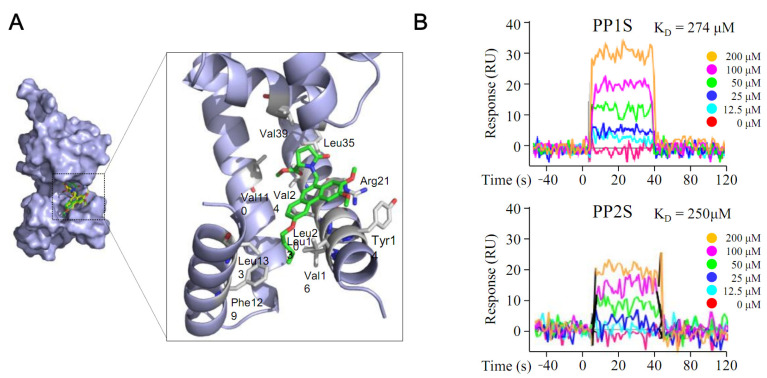
Interaction between PPs and Rv2887 protein. The interaction between PPs and Rv2887 protein was confirmed by (**A**) in silico and (**B**) SPR experiments. As the PPs increased to 200 μM, the response value increased concentration-dependently. Here, PP1S and PP2S have molecular weights of 423 and 465, respectively, and 200 μM represents 84.6 and 93.0 μg/mL, respectively.

**Table 1 antibiotics-11-01349-t001:** Effectiveness of PPs in combination with first-line anti-tubercular drugs.

Organism ^a^	Antibiotic ^b^ Combination	FIC_A_ ^c^	FIC_B_ ^c^	FIC ^d^
*M. tuberculosis* H37Rv	PP1S + INH	1	1	2
	PP1S + RIF	0.125	1	1.125
	PP1S + STR	1	0.125	1.125
	PP1S + PZA	1	1	2
	PP1S + EMB	1	1	2
	PP2S + INH	0.5	1	1.5
	PP2S + RIF	0.125	1	1.125
	PP2S + STR	1	0.125	1.125
	PP2S + PZA	0.5	1	1.5
	PP2S + EMB	1	1	2
XDR *M. tuberculosis*	PP1S + INH	0.06	1	1.06
	PP1S + RIF	0.5	1	1.5
	PP1S + STR	0.125	1	1.125
	PP1S + PZA	0.5	1	1.5
	PP1S + EMB	1	1	2
	PP2S + INH	0.25	1	1.25
	PP2S + RIF	1	1	2
	PP2S + STR	1	0.06	1.06
	PP2S + PZA	1	1	2
	PP2S + EMB	0.5	1	1.5

^a^ Combinations were tested against *M. tuberculosis* H37Rv or XDR *M. tuberculosis*. ^b^ INH, isoniazid; RIF, rifampicin; STR, streptomycin; EMB, ethambutol; PZA, pyrazinamide. ^c^ FIC_A_, MIC_A_ (MIC in drug A alone) in combination with drug B divided by MIC_A_; FIC_B_, MIC_B_ in combination with drug A divided by MIC_B_. ^d^ FIC = FIC_A_ + FIC_B_ (≤0.5, synergistic; 0.5–0.75, partial synergistic; 0.75–1, additive; 1–4, indifference; >4, antagonism).

## Data Availability

The data presented in this study are available on request from the corresponding author.

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
