# Peer review of "Increased Susceptibility of *Mycobacterium tuberculosis* to Ethionamide by Expressing PPs-Induced Rv0560c"

_antibiotics, 2022, doi:10.3390/antibiotics11101349_

Round 1

Reviewer 1 Report

In the current manuscript by Song. Ho-Y. and co-workers have claimed the sensitivity of ethionamide (ETH) to M. tuberculosis increases due to PP-induced overexpression of Rv0560c. The PPs (methyl (S)-1-((3-alkoxy- 6,7-dimethoxyphenanthren-9-yl) methyl)-5-oxopyrrolidine-2-carboxylate derivatives) were previously reported by the same research group as anti-tuberculosis agent. Their 2-dimensional electrophoresis experiment combined with microarray and RT-PCR experiments showed a selective increase in Rv0560c protein expression. Further, the protein microarray experiment for all proteins of M. tuberculosis revealed the highest interaction for the EthR-Rv0560c protein pair. They observed a synergistic effect of 0.5x MIC of PPs on ETH-mediated M. tuberculosis H37Rv growth inhibition. The in silico analysis combined with the SPR experiment suggested PPS1 and PPS2 interaction with Rv2887 protein, a repressor of Rv0560c.

All though the authors had previously reported the PPs as an effective anti-tuberculosis agent and it is important to comprehend the PPs mechanism of action in M. tuberculosis growth inhibition. The current manuscript has tried to investigate the PPs mode of action and come up with a few conclusions. But the derived conclusions in the manuscript are not well supported by experimental evidence, the authors need to provide more experimental evidence to support their proposed hypotheses/claims. For example, 

1)    The observed synergistic effect of PPs on ETH-mediated M. tuberculosis H37Rv growth inhibition is proposed due to the EthR-Rv0560c interaction. But there is no evidence presented here to show the EthR-Rv0560c interaction has any direct inhibitory effect on the EthR mediated EthA (ETH bio activator) repression. More experimental evidence requires to support this claim.

2)    They concluded that the PPs interaction with Rv2887 is responsible for the higher overexpression of Rv0560c protein. But again, there is no direct evidence presented here to show that the PPs-Rv2887 interaction inhibits the Rv2887 mediated Rv0560c regulation. The authors should provide structural or biochemical evidence suggesting PPs-Rv2887 binding is important for Rv2887 gene regulation function.

Overall, I would recommend the current manuscript for major revision. The authors should provide more comprehensive and direct experimental evidence to support their conclusions and claims.

Author Response

Response to Reviewer 1 Comments

Proposed Revisions:

In the current manuscript by Song. Ho-Y. and co-workers have claimed the sensitivity of ethionamide (ETH) to M. tuberculosis increases due to PP-induced overexpression of Rv0560c. The PPs (methyl (S)-1-((3-alkoxy- 6,7-dimethoxyphenanthren-9-yl) methyl)-5-oxopyrrolidine-2-carboxylate derivatives) were previously reported by the same research group as anti-tuberculosis agent. Their 2-dimensional electrophoresis experiment combined with microarray and RT-PCR experiments showed a selective increase in Rv0560c protein expression. Further, the protein microarray experiment for all proteins of M. tuberculosis revealed the highest interaction for the EthR-Rv0560c protein pair. They observed a synergistic effect of 0.5x MIC of PPs on ETH-mediated M. tuberculosis H37Rv growth inhibition. The in silico analysis combined with the SPR experiment suggested PPS1 and PPS2 interaction with Rv2887 protein, a repressor of Rv0560c.

All though the authors had previously reported the PPs as an effective anti-tuberculosis agent and it is important to comprehend the PPs mechanism of action in M. tuberculosis growth inhibition. The current manuscript has tried to investigate the PPs mode of action and come up with a few conclusions. But the derived conclusions in the manuscript are not well supported by experimental evidence, the authors need to provide more experimental evidence to support their proposed hypotheses/claims. For example, 

1)    The observed synergistic effect of PPs on ETH-mediated M. tuberculosis H37Rv growth inhibition is proposed due to the EthR-Rv0560c interaction. But there is no evidence presented here to show the EthR-Rv0560c interaction has any direct inhibitory effect on the EthR mediated EthA (ETH bio activator) repression. More experimental evidence requires to support this claim.

2)    They concluded that the PPs interaction with Rv2887 is responsible for the higher overexpression of Rv0560c protein. But again, there is no direct evidence presented here to show that the PPs-Rv2887 interaction inhibits the Rv2887 mediated Rv0560c regulation. The authors should provide structural or biochemical evidence suggesting PPs-Rv2887 binding is important for Rv2887 gene regulation function.

Overall, I would recommend the current manuscript for major revision. The authors should provide more comprehensive and direct experimental evidence to support their conclusions and claims.

Response:

We want to thank you for pointing out the logical limitations that are very important in this study.

1) As mentioned in the manuscript, in ETH's activation by EthA, how EthR acts as a repressor for EthA has already been well elucidated. With this mechanism in mind, the observed interaction between Rv0560c overexpressed by PPs and EthR and the result of increased sensitivity of PPs to ETH was judged based on the EthR-EthA pathway mechanism. However, as the reviewer pointed out, I think there is a limit to the fact that direct experimental results do not fully support it.

2) In addition, the function of Rv2887 as a repressor for Rv0560c has been previously reported. This study confirmed the interaction between PPs and Rv2887 and the result of overexpression of Rv0560c. Nevertheless, we agree with the reviewer's comment that more experiments are required to determine that this result is also due to the Rv2887-Rv0560c pathway.

  Thanks again to the reviewers for making critical comments. Moreover, these limitations are so important that they are newly described in the fourth paragraph of the discussion.

  This limitation will be investigated in more detail in future studies, and we would like to thank the reviewers again for their help in setting the research direction of this research team.

Reviewer 2 Report

The manuscript antibiotics-1935178 reports experimental data on the effect of PPs using Mycobacterium tuberculosis as experimental organism.

The topic is extremely interesting, the results are new and increase the knowledge in the field. The experimental plan is in principle correct, but the methodological approach is not adequately described (and needs to be corrected and implemented).

The manuscript is well organized and well written. I found only some typos in the text, easily solved with a quick revision. The discussion is sufficiently argued.

I consider that this manuscript is appropriate for publication in Antibiotics, and I suggest accepting it after minor revision.

1) I appreciate concise introductions, but this one is very short indeed. I suggest the authors to add some more content to better contextualize the research carried out. The final part, then, is more or less a summary of what has been carried out, while the research objectives should be more clearly explained, highlighting the novelties with respect to experiments carried out previously.

2) For the analysis of mRNA and protein expression, how was the statistical significance of the differences between the different experimental groups assessed? How many technical and experimental replicas have been made? What do the error bars represent in the graphs? The standard deviation or the standard error? Authors should add details of the statistical analysis performed (for example the software used) to the Materials and Methods and report statistically significant differences in the graphs.

3) Materials and methods should be more detailed. For example, the companies supplying the material used should be cited according to the journal's rules. How was the achievement of the intermediate exponential growth stage evaluated? By visible spectrophotometry? At what wavelength? At what speed and temperature was the cell colure centrifuged to obtain the cell pellet? How were the integrity and concentration of the extracted RNA verified? What procedure and materials were used for cDNA synthesis? What method was used to extract the proteins? What is the method used for the synthesis of the 3 proteins of interest?

4) The Conclusions are only 4 lines. I suggest merging them into the Discussion section.

Author Response

Response to Reviewer 2 Comments

Proposed Revisions:

The manuscript antibiotics-1935178 reports experimental data on the effect of PPs using Mycobacterium tuberculosis as experimental organism.

The topic is extremely interesting, the results are new and increase the knowledge in the field. The experimental plan is in principle correct, but the methodological approach is not adequately described (and needs to be corrected and implemented).

The manuscript is well organized and well written. I found only some typos in the text, easily solved with a quick revision. The discussion is sufficiently argued.

I consider that this manuscript is appropriate for publication in Antibiotics, and I suggest accepting it after minor revision.

1) I appreciate concise introductions, but this one is very short indeed. I suggest the authors to add some more content to better contextualize the research carried out. The final part, then, is more or less a summary of what has been carried out, while the research objectives should be more clearly explained, highlighting the novelties with respect to experiments carried out previously.

2) For the analysis of mRNA and protein expression, how was the statistical significance of the differences between the different experimental groups assessed? How many technical and experimental replicas have been made? What do the error bars represent in the graphs? The standard deviation or the standard error? Authors should add details of the statistical analysis performed (for example the software used) to the Materials and Methods and report statistically significant differences in the graphs.

3) Materials and methods should be more detailed. For example, the companies supplying the material used should be cited according to the journal's rules. How was the achievement of the intermediate exponential growth stage evaluated? By visible spectrophotometry? At what wavelength? At what speed and temperature was the cell colure centrifuged to obtain the cell pellet? How were the integrity and concentration of the extracted RNA verified? What procedure and materials were used for cDNA synthesis? What method was used to extract the proteins? What is the method used for the synthesis of the 3 proteins of interest?

4) The Conclusions are only 4 lines. I suggest merging them into the Discussion section.

Response:

1) Thanks for pointing out the lack in the introduction part. It was prepared to describe the necessary details briefly, but as the reviewer pointed out, it is considered insufficient to enhance the reader's understanding. Accordingly, the second paragraph of the introduction part of the manuscript, which was revised by referring to previous review papers on the history of tuberculosis drug development, was newly added. Moreover, the fourth paragraph of the introduction was described with novelty so that the reader could asume this study in advance. By doing this, the completeness of this manuscript was improved, and I would like to thank the reviewer who advised making this possible.

2),3) We agree with the reviewer's suggestion that materials and methods, including statistical methods, must be described more specifically. The reviewer particularly pointed out '4.3. Exploration of gene expression of M. tuberculosis changed by drugs' was described too briefly, so it was presented in more detail. In addition, information about the software used for statistical and data analysis was added to "4.4. Microarray" and "4.5. 2DE". However, it was a little burdensome to present all the detailed experimental methods in the manuscript, so exact methods for microarray and 2DE experiments were additionally offered in the supplementary material after the protein synthesis part.

(Microarray and 2DE methods were added following the previously presented synthesis of Rv3855, Rv2887, and Rv0560c proteins and 'rabbit polyclonal antisera Preparation.’)

Thank you once again to the reviewer for kindly pointing out the insufficiency of the experimental method.

We would be very grateful if you could review this part again.

4) As advised by the reviewer, we moved the conclusion to the last part of the discussion section.

We want to thank the reviewers again for your insightful comments and advice to improve the quality of this paper. We have prepared our responses as best as we can. If our answers are insufficient, please let us know, and we will try our best to improve them again.

Corrected text is indicated in green in the revised manuscript.

Reviewer 3 Report

This is very interesting paper. However, to ensure author's claim is acceptable, additional experiments should be added as below.

1. Please include chemical structure of PP1S and PP2S.

2. Summary figure should be included in Conclusion section.

3. CETSA is also required for the confirmation of drug's target (Rv2887) interation with some mutants at putative binding sites.

4. Statistical significance should be included in all data.

5. Additional data (eg., cytokine levels or histology, etc) should be included to support its synergistic activity of PPs in mouse model.

6. Please explain the different doses used in experiments (100 uM in Fig. 1D or 200 uM in Fig. 5B)

Author Response

Response to Reviewer 3 Comments

Proposed Revisions:

This is very interesting paper. However, to ensure author's claim is acceptable, additional experiments should be added as below.

  1. Please include chemical structure of PP1S and PP2S.
  2. Summary figure should be included in Conclusion section.
  3. CETSA is also required for the confirmation of drug's target (Rv2887) interation with some mutants at putative binding sites.
  4. Statistical significance should be included in all data.
  5. Additional data (eg., cytokine levels or histology, etc) should be included to support its synergistic activity of PPs in mouse model.
  6. Please explain the different doses used in experiments (100 uM in Fig. 1D or 200 uM in Fig. 5B)

Response:

1) The chemical structures of PP1S and PP2S are included with IUPAC names in Supplementary Material Figure S4.

2) The conclusion has been revised along with a summary diagram of the newly proposed 'Rv2887-Rv0560c-EthR-EthA pathway' in this study.

3) While conducting this study, we tried to validate it as a variety of studies that validate drug targets, but we couldn't do it this time. However, through this revision, we will try to apply the method the reviewer taught us to future research plans. The fourth paragraph of the discussion presents the limitations of this study and those pointed out by the reviewer.

4) The missing statistical significance descriptions were supplemented by additional explanations (Figure 3, Figure S2).

5) As can be seen from the reviewer's advice, we should have been able to support the conclusions from animal experiments by conducting various experiments, but so far, we have not been able to. This drug candidate was judged to have development value, and additional experiments were planned. At that time, according to the reviewer's advice, we will perform various experiments that can complement animal efficacy. This limitation is described in the last paragraph of the fourth paragraph of the discussion section.

6) This was to confirm that the response value increases in a concentration-dependent manner while increasing the concentration of PPs up to 200 μM, that is, about 100 μg/ml. Because the concentration unit is different, there may be confusion, so the converted concentration is additionally presented as an explanation for Figure 5.

Thank you very much for your careful review and advice.

We want to thank the reviewers again for your insightful comments and advice to improve the quality of this paper. We have prepared our responses as best as we can. If our answers are insufficient, please let us know, and we will try our best to improve them again.

Corrected text is indicated in green in the revised manuscript.

Round 2

Reviewer 1 Report

The manuscript has improved significantly after revision. I am satisfied with the author's response. I will recommend the manuscript for publication in its current form.

Reviewer 3 Report

This paper is now acceptable, since all issues are well explained.